# Information Communication Technology and Infant Mortality in Low-Income Countries: Empirical Study Using Panel Data Models

**DOI:** 10.3390/ijerph19127338

**Published:** 2022-06-15

**Authors:** Issam Khelfaoui, Yuantao Xie, Muhammad Hafeez, Danish Ahmed, Houssem Eddine Degha, Hicham Meskher

**Affiliations:** 1School of Insurance and Economics, University of International Business and Economics, Beijing 100029, China; xieyuantao@uibe.edu.cn; 2Institute of Business and Management Sciences, University of Agriculture, Faisalabad 38000, Pakistan; hafeez_86@hotmail.com; 3School of Finance and Economics, Jiangsu University, Zhenjiang 212013, China; danish.ahmed88@live.com; 4School of Foreign Language, Shanghai Jianqiao University, Shanghai 201315, China; 5Department of Business Administration, HANDS—Institute of Development Studies (HANDS-IDS), Karachi 75230, Pakistan; 6Center for Islamic Finance, University of Bolton, Bolton BL3 5AB, UK; 7International Institute on Governance and Strategy (IIGS), Beijing 100000, China; 8Department of Mathematics and Computer Science, Faculty of Science and Technology, Université de Ghardaia, Ghardaia 47000, Algeria; degha.houssem@outlook.com; 9Kasdi Merbah University, Ouargla 30000, Algeria; hicham.meskher@g.enp.edu.dz

**Keywords:** health, infant mortality, ICT, PMM, panel regression, low-income countries, Driscol–Kraay, instrumental variables IV, pooled common correlated effects PCCE

## Abstract

According to the World Health Organization, lower-income countries suffer from adverse health issues more than higher-income countries. Information and communication technologies (ICT) have the potential to resolve these issues. Previous research has analyzed the theoretical and empirical causal effects of ICT on infant mortality at country-specific and global levels for a short period of time. However, the causes and results could be different in low-income countries. The objective of this paper was to examine the deficiencies through the use of panel data from 27 low-income countries from 2000–2017. We applied the predictive mean matching technique to supplement the missing data and then used panel data techniques (i.e., fixed effects (FE) and pooled common correlated effects (PCCE)), and system-GMM to estimate the causal effects. We compared the consistency and the possible heterogeneity of previous results using a set of robust techniques and empirical tests. We found that internet access and, to a lesser extent, cellular mobile subscriptions, two of the three ICT variables used in our research, had a significant positive effect on reducing infant mortality in low-income countries. In conclusion, governments and policymakers of low-income countries should consider the availability of internet-related ICT innovations and make them nationally accessible to reduce health crises such as the infant mortality rate.

## 1. Introduction

“A disease threat anywhere is a disease threat everywhere” is a quote from the Centers for Disease Control [1]. The global COVID-19 pandemic showed us that if health is affected anywhere, it will affect everywhere. Even though health outcomes such as life expectancy and child mortality rates are improving globally, the minimum life expectancy and maximum infant mortality rates are disappointing: 39.54 years old and 147.86 per thousand children, respectively [2,3]. Unfortunately, approximately 1 million newborns do not survive their first day, and 2.6 million do not survive their first month [4,5]. In addition, physical and mental health issues are increasing in low-income countries. This increase places a heavier burden on low-income economies and widens the gap between developing and developed countries.

To address this issue, economists first defined “health outcome or health issue” as the healthcare parameters that impact people’s lives and health in any given country. These parameters include longevity, the use of health-designated services, psychosocial and physical states, morbidity and mortality, costs of healthcare, and chronic and non-chronic illnesses [6]. Researchers have measured these parameters using indicators such as stillbirths, neonatal mortality, under-five child mortality, maternal mortality, total fertility, low birth weights in infants, life expectancy at birth, and sex at birth [2]. Finally, policies have been proposed to mitigate and reduce these health issues. For example, in 2015, the three top sustainable development goals (SDGs), out of seventeen goals, re-established by the U.N. General Assembly included the reduction and eradication of all forms of poverty in the world; comprehensive and fair education for all, and the improvement and preservation of well-being and health regardless of age [7].

Another example of proposed solutions has been the use of information and communication technologies (ICT). In this globalized epoch, ICT is a crucial factor for innovative infrastructure and is internationally recognized as a principal determinant for societies’ socio-economic growth and improvement. Many comprehensive studies have illustrated the role of ICT in increasing the rate of GDP, unemployment reduction, capacity improvement, economic extension, and expansion through fast and efficient information exchange, and poverty mitigation [8,9,10]. A concise definition of ICT by the World Bank was “the collection of electronic operations and norms that assist the acquisition, storage, management, transfer, and exhibition of information”. ICTs have improved the healthcare systems of developing economies [11]. ICT challenged the location and time limitations of information exchange with better performance at a lower cost [12]. ICT has supported health literacy and eased the attainment of health education. Moreover, researchers have shown strong evidence that ICT develops the quality of life as one of four main pathways to attain socio-economic growth [13].

However, the development, innovation, and advancement of technology for the purpose of improving health issues, such as infant mortality, have been insufficient [14]. Therefore, the goal should be less idealistic and more focused: one objective and one measure at a time. The applications, and their subsequent failures, have taught policymakers that technology and technological solutions have not advanced far enough to resolve serious health crises in low-income economies. Furthermore, low-income countries may not have the capabilities or specializations to employ even accessible ICT and other technological solutions due to government incompetence and/or corruption.

In our study, we focused on 27 low-income countries and examined the relationship between ICT and health. Using 18 years of data from the World Bank, from 2000 to 2017, we studied the effect of three ICT proxies, including internet use (IU), mobile cellular subscription (MCS), and fixed phone subscription (FTS), on health outcomes rates: infant mortality (INFM) and child mortality under five (CMU5). Our objectives included the following: (1) To separate and differentiate our study from the global scope as well as the global results and findings; (2) To determine if there are any significant positive or negative effects of ICT on health variables for low-income countries if they vary according to the proxy measurement and if they are significantly different from global predictions and estimates. (3) To provide policy suggestions based on the results. To the best of our knowledge, this was the first panel study on this subject for low-income countries over a long period of time.

## 2. The Literature on the Effects of ICT on Health Outcomes

The development of ICT and the internet has provided access to an abundance of health-related information for individuals. This accessibility has improved the effectiveness and efficiency of patients when communicating with their physicians, assisted them in their health decisions, and improved healthcare-related interactions and services [15,16,17,18].

ICT has supported independent living measures by facilitating easier communication strategies for the elderly, so they can contact their loved ones, family members, and physicians [19,20,21]. As indicated by Lacal and Mechael [22,23], due to the advances in health-relevant applications, ICT has enabled real-time advice and feedback in healthcare scenarios and assisted in decentralizing healthcare.

Bukachi and Pakenham-Walsh [24] found that healthcare professionals used the internet for correspondence and recording health-related data in a coordinated effort around the world. Wald et al. [25] and Bankole et al. [26] used survey data to explore the relationships among healthcare frameworks, internet access, and doctor–patient relationships. They found that internet access improved health frameworks and global health.

In developing economies, Lucas [27] examined the healthcare impact of ICT innovations such as the improvements in health information systems; computer-aided prescription monitoring, treatments, and diagnostics; a variety of applications for “telemedicine”; and educating large populations about their health and healthcare. In the same developing economies, Blaya et al. [28] reviewed the developments of e-health information technologies to manage patient care. They showed that e-health had a significant impact on health in developing countries and that ICT enhanced interactions among institutions, assisted in organizing and distributing medicines, and promoted monitoring and identifying patients who may not complete necessary treatments. Furthermore, personal mobile devices have increased both the time and quality of data acquisition. Cole-Lewis and Kershaw [29] presented a systematic review of mobile text messaging as an instrument to change individuals’ behavior toward disease management and illness prevention, and they found significant impacts on diabetes control, smoking cessation, and weight reduction. Their results showed that text messaging could be effective for practical disease prevention and control.

Moreover, ICT can provide cost-effective solutions for health improvement. For example, it empowers individuals affected by similar health challenges to share information and experiences [30]. Déglise et al. [31] reviewed contextual studies from developing countries to investigate SMS interventions for the prevention, monitoring, supervision, and treatment of communicable and non-communicable illnesses and diseases. They found that mobile devices played an effective role in disease control in developing nations as they are cost-effective and assist in mitigating the demands and deficiencies in health systems. Nilseng et al. [32] conducted a pilot study in 2010–2011 in Tanzania and found that ICT tools improved inventory and ordering management among healthcare operators. Fedha [33] suggested that information innovation significantly affected clinical appointment attendance among pregnant females, which could potentially improve the rate of infant mortality. 

In recent studies, ICT removed the geographical barriers between medical care suppliers and the recipients by improving their export potential [34]. Cole et al. [35] assessed the health advice provided by weblogs and online forums. Their examination indicated that there was little evidence indicating the information was erroneous, and they proposed that these sources allowed individuals to ask health-related questions and receive quality responses. Tsai et al. [36] investigated a blended e-learning system (BELS) used by Taiwanese nursing personnel and found that its e-learning curriculum included face-to-face courses in advanced health education topics. Majeed and Khan [3] conducted a panel data analysis in 148 countries during 1990–2014 to analyze the relationship between ICT and health outcomes. They validated that ICT increased global health outcomes in terms of prolonging life expectancy and lowering infant mortality rates, and they recommended healthcare plans and policies that focused on digital inclusion. Similarly, Dutta et al. [2] conducted a panel data analysis on 30 Asian countries from 2000 to 2014 to study the long-term impact of ICT on health outcomes. Their findings found a significant effect of ICT on healthcare. 

However, there have also been negative outcomes when using ICT solutions in healthcare. The quality of information collected by ICT may be of inferior quality, misleading, or easily misinterpreted, which can jeopardize health behaviors and outcomes. Kiley [37] indicated that false information could provoke fear and worry about preventable illnesses, or even death. In addition, there are still disparities in Internet access among different socioeconomic groups (rural versus urban, wealthy versus poor, etc.) even in developed countries. This problem on a global scale may increase the disparities between these groups, complicating an already complex issue. Third, easy access to online health information could result in an increase in unnecessary medical visits due to misinformation or patient misperceptions, which could place an undue burden on healthcare systems [38].

In [39,40,41], the researchers suggested that low-income countries should focus on improving their health infrastructures and access to clean water, as opposed to developing ICT tools. Tanis et al. [42] found that health anxiety motivated patients to seek out health information online, yet the same motivation was negatively associated with their satisfaction with their medical consultation.

The main barrier to employing ICT and innovations are its implementation, especially in developing countries. In addition to the lack of skilled workers and specialists, the economic costs may be too high; these can include licensing charges, yearly or monthly paid upgrades, subscription fees, and replacement expenses. Furthermore, in the long term, the literature has indicated that significant reliance on foreign resources creates yet another dependency cost. Moreover, the harsh climate of various developing countries can damage ICT hardware and instruments that require climate-controlled settings and dust-free conditions. In recent years, policymakers and researchers have examined the health outcomes with ICT and the related infrastructure. However, there have been opposing theoretical results, especially in low-income countries. Therefore, an empirical evaluation of the link between health and ICT specifically in low-income countries is necessary for a comprehensive understanding of the relationship. Previous research has been centered on clinical and/or country-specific data that cannot be generalized to various samples. In our study, we empirically tested the assumed relationship of ICT tools with health outcomes in 27 low-income countries over 18 years.

## 3. Materials and Methods

### 3.1. Data Source, Data Processing, and Data Imputing

#### 3.1.1. Data Source

In Table A1, we provide a brief summary of the variables used and their sources in previous research. We only considered works that studied the impacts of ICT and its development on health outcomes in national panel studies (i.e., studies that used aggregated datasets). In accordance with previous studies, we acquired the necessary data from the World Bank database. The World Bank defined low-income countries as nations that have a per capita gross national income (GNI) of less than USD 1026. Twenty-eight countries were identified using this criterion: Afghanistan; Burundi; Benin; Burkina Faso; Central African Republic; Congo, Democratic Republic; Eritrea; Ethiopia; Guinea; Haiti; Liberia; Madagascar; Mali; Mozambique; Malawi; Niger; Nepal; Korea Democratic People’s Republic; Rwanda; Sierra Leone; Somalia; Syrian Arab Republic; Chad; Togo; Tajikistan; Tanzania; Uganda; and Yemen Republic.

#### 3.1.2. Data Processing

To process the data, we conducted five steps:

Step 1. Variable selection: In this initial step, we favored the variables that were mentioned in previous literature.

Step 2. Time period: To determine the time period used in our study, we relied on data availability. Since both the data before 2000 and after 2017 were not available for our variables of interest (ICT) or missing, we selected a period from 2000 to 2017.

Step 3. Variable exclusion: We excluded variables that were completely or partially (20% missing) absent in the data for 28 countries, which were assigned by excluded in the last column of Table A1.

Step 4. Country exclusion. We excluded countries with missing variables or unrealistic values for our variables of interest, specifically North Korea.

Step 5: Data replacement. Missing data with a rate of 7.5% was estimated using the predictive mean matching (PMM) technique. The strength of any imputation technique is greater with more information provided. We only included three variables (the prevalence of HIV index, life expectancy at birth index, and incidence of tuberculosis index) in this step, but we excluded them from the analysis.

Overall, we had 27 low-income country data for 18 years with the included variables, as detailed in the fifth column of Table A1.

#### 3.1.3. Predictive Mean Matching (PMM)

As mentioned previously, to complete the small percentage (7.5%) of our dataset that were missing values, we applied the standard PMM technique. PMM calculates missing continuous values in datasets. PMM attributes a present observed value that is nearest to the missing one and preserves the initial distribution of the experimental data better than fully parametric multiple imputation strategies and is known to be consistent and robust for a small percentage of missing values [43,44]. 

### 3.2. Empirical Tests

To utilize the model described beforehand and to evaluate the robustness of the results, we used a set of tests. The tests we used, which have been widely adopted in the literature, were: the panel unit-root test, panel causality tests, pool-ability tests, panel cross-dependency tests, and Hausman specification test. A brief definition of the test is provided in Section A.1 empirical tests part.

### 3.3. Model Construction

In Grossman’s [45] health production function, health was indicated by individual comportment, medical care, and resources. Theoretically, Grossman’s production of health function is:(1)Health=f(health inputs)
where “Health” is a person’s output of health, and health inputs are the following factors: income and wealth, education level, health expenses and investments, health accommodations, milieu, and the standard of living. Grossman’s [45] model studied the health production function at a micro-level. Fayissa and Gutema [46] converted it to its macro equivalent. They re-expressed health inputs per capita and reconstructed them in three divisions: economic, environmental, and social.
(2)Health=f(ECO, ENV, SOC)
where ECO, ENV, and SOC describe factors of economic, environmental, and social variables, respectively. There are numerous variables under each division. Therefore, every researcher has adopted different variables due to data and resource limitations as well as other reasons.

In our empirical research, the economic factors were economic growth (GDP and ADR) and health expenditures (HE, IM, and IDPT). The social variables were education (SCH) and sex ratio (SR), and the environmental factors included urban population ratio (UP) and carbon dioxide emissions (CO2E).
(3)Health=f(Economic growth, Health expenditures, SCH, SR, UP, CO2E)

Our research explored other potential factors that could affect health by concentrating on ICT. To consider the effect of ICT on the production of health models, Majeed and Khan’s [3] prolonged Equation (3) to incorporate digital inclusion is as follows:(4)Health=f(Economic growth, Health expenditures, SCH, SR, UP, CO2E, ICT)

In this study, we measured ICT through fixed telephone subscriptions (FTS), mobile cellular subscriptions (MCS), and internet use (IU). As follows, we quantified health through two infant mortality proxies (INFM and CMU5).

According to [2,3], the above equation between ICT and Health and our model could be formulated to the following panel equation design by taking into consideration the ease to build a link between them through using log functional form and to interpret the coefficients.
(5)$$lnHealth_{it}=\alpha_{it}+\beta_{1}lnICT_{it}+\beta_{2}lnX_{it}+\varepsilon_{it}\;$$
where i= country index, and t= time yearly index. Health represents the dependent variable in our design measured by INFM and CMU5. ICT represents the independent variable in our design measured by IU, FTS, and MCS. X represents the set of control social, economic, and environmental factors mentioned before and measured by GDP, SCH, SR, IM, IDPT, HE, ADR, UP, CO2E. ln is the natural logarithm function, and  ε indicates the idiosyncratic error term. To estimate the panel model parameters, we first applied the pooled ordinary least squared (POLS) and then the random and fixed-effects models. The variable $\alpha_{i}$ has three forms: (1) $\alpha_{0}$ constant for the POLS model; (2) $\alpha_{i}$, $\alpha_{t}$, and $\alpha_{it}$ country, time, and two-way (country and time) fixed effects FE, respectively, in the fixed effects model; (3) $\alpha_{0}+\alpha_{it}$ the constant and random effect errors, respectively, in the random effect model. To address the cross-sectional dependence, we estimated the three regression models (POLS, random, fixed effects, RE/FE) with Driscoll–Kraay standard errors. In the presence of cross-sectional dependence, we found that the standard errors of Driscoll–Kraay were well-calibrated [47,48]. To address the strong observed and unobserved cross-dependence in the data including “when the regressors and errors both have a factor structure”, we applied common correlated effects (CCE) modeling. This approach has become very influential in both the theoretical panel-data literature and in empirical applications. The CCE method, which was presented by Pesaran [49] and developed by many authors including [50], consists of an equation of interest, in which the error has a component structure, and a reduced form equation, in which the explanatory variables are linear functions of the same factors that present in the main equation. CCE then treats the response and explanatory variable cross-sectional averages as fixed effects, removing unobserved variability. In the latest paper presented by Juodis et al. [51], they provided strong evidence of the robustness of the pooled CCE. Therefore, in the present research, we also applied the pooled PCCE with Driscoll–Kraay robust errors. The estimation of linear panel data models using generalized least squares (GLS) was also used. In the presence of heteroskedasticity and serial and cross-sectional correlations, the proposed GLS estimator outperformed OLS. Finally, three factors were likely to cause endogeneity in our model: (1) simultaneous links between health and ICT factors, (2) ICT measures found to correlate with error terms, and (3) omitted variable bias. These issues could be mitigated by the use of instrumental variables. To account for potential endogeneity, this study employed a two-stage least squares in FE and system-GMM in panel data. The potential endogenous variable (ICT) was adjusted with a variety of appropriate internal and external instruments. Initial values, communications, computers, and so on (% of service exports, BoP), and internet country code dummy were used as instruments. The indicators of communications, computers, and so on (% of service exports, BoP), and internet country code dummy were highly correlated with ICT measures. Furthermore, these indicators had no direct influence on population health. The data for instrument variables are detailed in Table A1 instrument variables.

We followed an algorithmic structure of data modeling to explain the use of these analyses. The algorithm is as follows:Ensure stationarity through the tests of 1st-generation panel unit-root;Test for pool-ability to utilize the POLS and, later, the PCCE;First step: Apply the POLS analysis and fixed effects to get their estimates results;Compare through F-test of individual effects to decide which estimates better depict our data and model;Second step: Apply random-effects analysis to obtain its estimates results;Compare RE with FE to decide which estimates depict better our model;Test for cross-sectional dependence through the results of the estimate of RE and FE;Apply appropriate techniques to remedy the presence of cross-sectional dependence;Add additional analyses to compare the final model estimates (robustness);To avoid redundancy and length, we only report the adequate results.

## 4. Results

In this research, we reported our results accordingly. The first subsection provides a broad description of the data, their Spearman correlations, unit-root test results, and panel Granger causality test results. Section 2, Section 3 and Section 4 report individual/two-way fixed effects with/without IV, PCCE, and system-GMM estimation results, respectively.

### 4.1. Descriptive Statistics

Table 1 provides the descriptive statistics of our variables. After reviewing the mean and the min–max of the dependent variables, we found that the results were high for low-income countries. On average, these countries lost 6.7% of their children at birth and 10.2% under five years old. In some country cases, the maximum was 14.2% and 23.4%, respectively. Throughout the years, both infant mortality rates decreased at a slow pace, see Figure 1b,d. In 18 years, the average lessened from 9% to 5% in infant mortality and from 15% to 7% in children under five. In Figure 1a,b, the averages vastly differed between countries, and except for Syria, most of the countries’ averages were above 5% for both infant mortality indicators. For Syria, we did not have a confirmed source; however, we speculated that Syria should be at the same level as a result of the ongoing civil war that started in 2011.

Similarly, Figure 2 and Table 1 show the changes on average in our independent ICT variables. The average was 4% of the total population that had internet access among low-income countries. The 4.5% average was the across-time-and-country average. The trend of individual access to the internet began approximately 20 years ago; therefore, Figure 2b shows the most recent averages. Panel (b) of Figure 2 shows that, in 2018, the average internet users were around 15% of the low-income countries’ populations. For FTS and MCS, the averages were 1.5% and 28.7%, respectively. Except for FTS, in Figure 2b,d,f, IU and MCS increased over time. Of the ICT variables, MCS showed the most growth over time, and from 2016 to 2018, it averaged approximately 60% of the total population, see Figure 2f. Figure 2a,c,e also show that the average user, for all ICT indicators, per year, differed from country to country, especially for IU and MCS.

Figure 1 and Figure 2 show the cross-country heterogeneity and dependencies for all variables, dependent and independent. This type of heterogeneity may have biased the results given by the estimation methods such as the standard POLS. In Appendix A, Figure A1 presents Spearman’s correlation matrices for our set of variables. In this research, we used these correlation matrices to indicate the possible causality between our sets of variables. The correlation matrices showed that there may have been a negative causal relationship. In other words, ICT scales and health outcomes presented through infant mortality indices correlated negatively. Therefore, health outcomes may have been positively affected by ICT. We also observed, as shown in Figure A1, that both INFM indices correlated negatively with other control variables, except for ADR and UP. Considering previous studies and the findings in Figure A1, we could estimate the causality direction between health/health input variables and ICT. 

Finally, in Table A2, the first-generation test columns, we report the LLC and IPS panel unit-root test results. We observed, by using the LLC method, that almost all variables were stationary at level except ADR and CO2E. Using the IPS method, not all variables were stationary at level. Even though some variables were stationary, we used the sum of total stationarity as an indicator to use static methods. While the dependent variables were stationary at level, only two of the three ICT variables were not stationary according to the IPS method. However, in future studies, the use of dynamic models would be prudent. We established the stationarity proposition. After establishing the cross-sectional dependencies, we also reported the test results of second-generation tests (see Table A2 s-generation panel unit-root tests). CIPS and CADF tests did not reject the assumption of the presence of cross-sections in the data. Similar to the results of the first-generation tests, we found that almost all the results rejected the hypothesis of the presence of unit-roots. Therefore, our panel was stationary with “drift” or “trend” settings. 

Finally, the last set of tests that we applied were the panel Granger causality tests. As we noted in Table 2, all the test results reported a significance causal impact of ICT indicators on both CMU5 and INFM.

### 4.2. Fixed Effects FE Results

In Table 3, in the second-to-the-last row, we found that the Breach–Pagan test results rejected the null hypothesis of an unbalanced dataset for the 1–8 model specification. As a consequence of the test results, we used the POLS.

Correspondingly, Table 3 reports the results of the estimation of FE estimators using robust Driscoll–Kraay and the cross-dependence test results, and the Hausman specification test results. The FE with Driscoll–Kraay robust error results were reported in eight columns. In columns 1–8, we noticed that the IU and MCS estimators were associated with a decrease in both INFM and CMU5. More specifically, in columns 1–3 and 5–7, INFM and CMU5 decreased by 7% and 6%, respectively, in association with IU. MCS reduces INFM and CMU5 by 6% and 5%, respectively. While in columns 4 and 8, where we had regressed all the ICT variables simultaneously, IU decreased INFM and CMU5 by 4% and 5%, respectively; MCS reduced both INFM and CMU5 by 3%. Hence, in the FE estimation method, FTS was not statistically significant. As per the other control variables, in Table 3, we observed that most were statistically significant. Except for CO2E, their significance direction or sign was similar to those in previous research. The last row in Table 3 shows the results of the Hausman specification test. The results of the test rejected the null hypothesis. Therefore, the FE estimates were more consistent than the RE estimates. The second- and third-to-last two rows display the cross-dependence test results. Breush-Pagan LM and Pasaran CD cross-sectional results rejected the null hypothesis. These results indicate that the FE estimation results without applying Driscoll–Kraay robust errors did not account for country cross-dependence as well. As per the results of FE with instrumental variables (IV), we found the only ICT variable that underwent a change in its estimate. In Section B.1 Table A3, we found that IU decreased INFM and CMU5 by 18% and 14%, respectively. We did not apply the IV for all models simultaneously as the number of regressors was larger than the number of IVs.

### 4.3. Two-Way FE Using Driscoll–Kraay Robust Errors with IV Estimation Results

Similarly, Table 4 shows the results of the estimation of the two-way FE estimators using robust Driscoll–Kraay. The two-way FE accounted for both country and time effects. In Table 3, FE with Driscoll–Kraay robust error results are shown in eight columns, from column 1 to column 8. The estimates of ICT indicators were all negative. However, only MCS was statistically significant. IU and FTS decreased both INFM and CMU5. It also indicated that MCS decreases both INFM and CMU5 significantly. More specifically, MCS reduced INFM and CMU5 by 3% and 2%, respectively. In the columns where we regressed all three variables simultaneously, MCS reduced both INFM and CMU5 by 2%. For other control variables, only GDP, IM, IDPT, and SR remained statistically significant, as shown in Table 4. The other control variables were not statistically significant. With the exception of HE, the estimates were statically significant for columns 6–8. One reason for the change in the results, especially for the control variables, was that the two-way estimation approach may have been a better fit to model our data. A more apparent indication for the previous proposition was that the total sum of squares and the residual sum of squares R-squared and adjusted-R-squared estimates were lower than the one-way (individual/country) FE results. In addition, the figures in our study showed an apparent time effect.

For the results of FE with IV, the estimation results of ICT were the same as the o individual/country effect, as shown in Table A3. In Table A4, the IU reduced INFM and CMU5 by (2% and 1%) and (3% and 2%), respectively. For the control variables, their estimation results were similar to those of the two-way effect of Table 4.

### 4.4. Panel Common Correlated Effects PCCE Estimation Results

Similarly, Table 5 reports the results of the estimation of PCCE estimators using robust Driscoll–Kraay. PCCE consists of an equation of interest in which the error has a component structure and a reduced form equation in which the explanatory variables are linear functions of the same factors that are present in the main equation. CCE then treats the response and explanatory variable cross-sectional averages as fixed effects, removing unobserved variability. Similar to FE Table 3 and Table 4, PCCE results were reported in eight columns, from column 1 to column 8. In the ICT individual models (columns 1–3 and 5–7), only the IU estimate was statistically significant. The IU decreased both INFM and CMU5 significantly, yet the significance level was low. In columns 4 and 8, where we regressed all three variables simultaneously, FTS significantly reduced both INFM and CMU5 by 1% while MCS increased CMU5 significantly. For the other control variables, in Table 5, we observed changes in estimates and significance levels. For example, GDP was statistically significant only in columns 5–7. SCH was statistically significant only in columns 4–8. IM, IDPT, HE, ADR, and UP were statistically significant for all the columns, except (3 and 5), (3 and 6), (1 and 5), (3 and 5), and (1,2, and 5), respectively. SR was statistically significant only in column 4. CO2E was statistically significant in all the columns. One reason for the change in the results, especially for the control variables, could have been that the PCCE estimation approach was not an ideal fit to model our data. The total sum of squares and the residual sum of squares R-squared and adjusted R-squared estimates were higher than the two-way FE models’ results, as well. 

### 4.5. System-GMM Results

Finally, the last approach that we applied, as reported in Table 6, was system-GMM. We used IV 2SLS to address potential endogeneity. Nevertheless, this approach was improper in the existence of heteroscedasticity. In this circumstance, it would be reasonable to utilize the generalized method of moments to address both heteroskedasticity and endogeneity. We employed the system-GMM presented by Arellano and Bond. In our process of estimating, we used an inverse matrix to keep all system-GMM estimators consistent in terms of heteroskedasticity. System-GMM is used to address cross-dependency and hidden heterogeneity in a model. It uses internal and external instruments to calculate our independent variables. Therefore, in our experience, it was the most suitable of the previous models to estimate the effect of ICT on health through INFM and CMU5. In the system-GMM, the Sargen test analyzed the overall validity of the instruments, the Arellano–Bond test for AR (1) and AR (2) examined the absence of the first-order and the second-order serial correlation in disturbances, and the Wald test was conducted for the coefficients. In Table 6, we reported the estimates and test results of the system-GMM of all the models, from 1 to 6.

First, we found that the Sargan test did not reject the null hypothesis, so the instruments were highly valid and exogenous. Second, the Arellano–Bond results did not reject the null hypothesis, indicating an absence of the first-order and second-order serial correlation in disturbances. Third, the Wald results rejected the null hypothesis. As a consequence, the estimates of this approach were valid. Fourth, we observed many changes in the estimated statistical significance and significance level, as compared to the previous methods. For example, in Table 6, we observed that the IU estimate was the only ICT estimate that had remained significant out of the three ICT estimates. In Table 6 columns 1–6, the IU estimate increased to 9.3% and 10.5% for both INFM and CMU5 models, respectively. The GDP, ADR, SR, UP, and CO2E estimates were statistically significant; SCH was only statistically significant in columns 3–6 while IM, IDPT, and HE were statistically significant.

## 5. Discussion

This study examined the impact of three ICT indices on health measured by INFM and CMU5. To achieve this aim, we applied suitable panel estimation approaches to mitigate any potential ICT and error heterogeneity and cross-sectional dependencies. We built our study model according to that of Grossman in 1972, including social, economic, and environmental factors, which we used for our health inputs. We used panel data of our variables for 27 low-income countries for 18 years (2000–2017). We found a strong cross-sectional dependency in our panel models and data. Before the application of our models, we confirmed the stationarity of our data through both generations of panel unit-root tests. Based on these tests, we noticed that the setting of our data differed from previous similar research: our data were stationary and cross-sectionally dependent. To remedy this difference, we used different estimation methods for the context of our data (i.e., Driscoll–Kraay robust estimates, common correlated effects, and generalized least squares). We also applied panel Granger causality tests to establish causality. Table 3, Table 4, Table 5 and Table 6 show only the results of the estimation methods (i.e., FE, two-way FE, PCCE, and system-GMM). All tables were divided into two parts: the first half shows INFM estimation results, and the second half shows CMU5 estimation results. Similarly, Appendix B
Table A3 and Table A4 show the results of FE with IV. Therefore, we divided our discussion into two parts as well:

### 5.1. INFM

We compared our results with similar research using OLS estimators. Majeed and Khan [3] employed a set of PLOS for their data from 184 countries. Dutta et al. [2] used fully modified and dynamic OLS analyses for the data from 30 Asian countries to study the effect of ICT on INFM. They found significant negative effects of the ICT indices on INFM. Comparatively, we found that some of our ICT variables had similar results. Our FE (both individual and country and two-way effect with/without IVs) and RE results showed that IU and MCS had significant effects on INFM, and the FTS effect on INFM was negative but not significant. Except for FTS, our results were similar to Lee’s [52] FE results and Majeed and Khan’s [3] FE and RE results. 

In our PCCE models, we found differences in the results, as compared to previous studies. Previous studies indicated that only individual IU had a significant negative effect on INFM. However, in the joint-variable model, FTS had a significant negative effect on INFM. Our system-GMM results differed from other studies. In recent research, only Lee [52] used the system-GMM in their dynamic panel model to study the effect of ICT variables on INFM. Using system-GMM, we found that only IU had a significant positive impact on reducing INFM. Similarly, Lee [52] found that both MCS and FTS had significant negative effects on INFM. 

### 5.2. CMU5 

For this health variable linked to ICT, the only research that included it in their design was Lee’s research in 2014. All our FE results were similar. In summary, all the findings via FE showed that IU and MCS have significant negative effects on CMU5, and there was no significant effect of FTS on CMU5. In the PCCE models, we found that our results were also different from those found in previous research: only IU when estimated individually had a significant negative effect on CMU5. However, in the joint-variables model, FTS (MCS) had a significant negative (net-positive) effect on CMU5. Again, the system-GMM results differed from Lee’s results [52]. While Lee [52] found a significant negative impact of only MCS on CMU5, we found a significant negative effect only of IU on CMU5.

In this paper, we discussed and empirically addressed the issues surrounding ICT and health in low-income countries. The FE (both individual and country and two-way effect with/without IVs) results suggested that mobile devices and internet access played important roles in reducing infant and child mortality rates, thus enhancing low-income countries’ health outcomes, while the usage of fixed phones is becoming obsolete and antiquated. The PCCE results suggested that IU, when considered individually, had a significant effect on reducing the INFM rate. Fixed telephones, in the presence of the other ICT variables, had a slight effect in reducing INFM. When considered with other ICT variables, mobile devices slightly increased the incidence of INFM. More precisely, the system-GMM showed strong evidence of the role of internet access. In the system-GMM results, in contrast to previous research, we found that for low-income economies, IU significantly reduced both child mortality rates, thus enhancing the health variables. This research was limited to many aspects that may provide direction for future research.

Our research had several limitations. First, to measure health outcomes, we used INFM proxies, which only included non-chronic diseases; future research could include chronic diseases. INFM, as we mentioned in the introduction, only considered the mortality rate in children, and it did not consider their future quality of life. Therefore, a better measure of health outcomes should be constructed. Second, the data in this research was also limited to the healthcare provided to the public sector. Future studies should consider more comprehensive panel data that include measurements and life quality as well as the private sector. Third, there are many cross-sectional country-specific studies on the adverse effects of the internet and the other ICT indices on mental health, such as increased stress, depression, and anxiety. This study did not address these issues. Consequently, there is a need to study the effects of ICT on global mental health, not just physical health. More specifically, broader health variables that consider maternal health, chronic diseases, mental health, and the quality of life should be taken into consideration by future studies. Finally, we encountered limitations in the generalization of our results. First, in our econometric design, we found a strong country cross-dependency. Second, the results were only applicable to cross-sectional analyses and approaches and would not be applicable for country-specific interpretation. Third, our analysis was not a dynamic design. Therefore, the results were static and interpreted as short-term dependencies. To solve this issue, we propose future research should use spatial dynamic panel models. To our knowledge, such a model was not yet accessible from a programming perspective and has only recently been introduced by Shi [53]. Shi’s [53] model, once programmed, may address hidden errors, variable heterogeneity, and cross-sectional dependencies. We would also suggest the use of a dynamic system-GMM and a difference GMM, with proper instruments, to address these design issues.

## 6. Conclusions

To conclude, our study expanded the current literature on health factors using an empirical analysis of ICT on health outcomes for 27 selected low-income countries for 18 years. We applied suitable panel estimation approaches to address potential error heterogeneity and cross-sectional dependencies (i.e., Driscoll–Kraay errors and PCCE). The health-outcome-dependent variables of this study were INFM and CMU5. The independent ICT proxies were IU, FTS, and MCS. In this study, we observed a strong presence of cross-sectional dependency in the models and data. Our panel data were stationary through both generations of panel unit-root tests. We established causality through panel Granger causality tests. In the FE results, we found that both IU and MCS decreased the rates of INFM and CMU5. In almost all the estimation results, we found that IU had a significant effect on the reduction in INFM and CMU5 rates. In the joint model of PCCE, fixed telephones may have also reduced both mortality rates while mobile devices appeared to increase them. Finally, due to the setting of our data and the application of novel models, our results differed from previous research and indicated that findings on a global scale were different from those found in low-income countries.

Our conclusions also provided policy suggestions for consideration. Policymakers and leaders of low-income countries should implement policies that guarantee sufficient internet access for the community. The internet, as a means of information transfer and a tool for communication, could improve these countries’ health outcomes by promoting health education, rapid health information sharing for early discovery and prevention of diseases, and healthcare systems support as well as overcoming the barriers of location and time for communication and consultation between patients and healthcare providers. Finally, and most importantly, the internet and mobile devices serve as great tools to inform and educate healthy behaviors, particularly for pregnant people, to reduce the child mortality rates through early contact with specialist physicians and healthcare systems, and to provide support and understanding for pregnancy and early childhood issues through the exchange of information.

## Figures and Tables

**Figure 1 ijerph-19-07338-f001:**
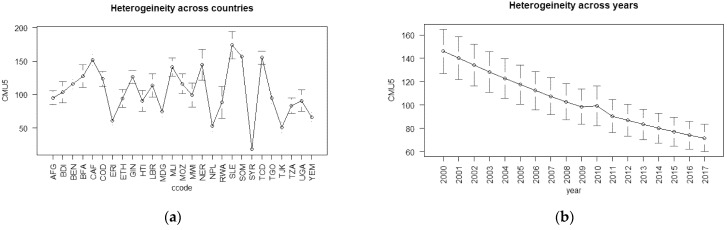
The time and year heterogeneity for the independent variables: CMU5 heterogeneity (**a**) across countries and (**b**) across years; INFM heterogeneity (**c**) across countries and (**d**) across years.

**Figure 2 ijerph-19-07338-f002:**
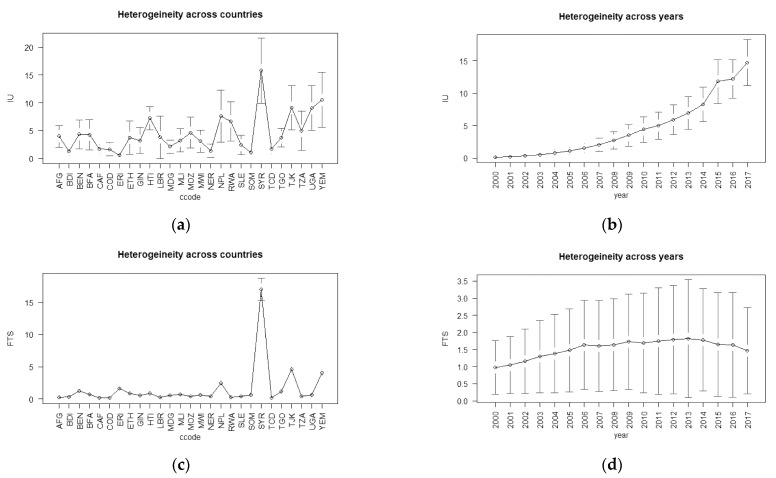
The time and year heterogeneity for the independent variables: IU heterogeneity (**a**) across countries and (**b**) across years; FTS heterogeneity (**c**) across countries and (**d**) across years; MCS heterogeneity (**e**) across countries and (**f**) across years.

**Table 1 ijerph-19-07338-t001:** Descriptive statistics.

Symbol	Descriptive Statistic				
	mean	SD	min	max	Median
Dependent Variables					
INFM	67.14	24.502	13.80	142.40	66.25
CMU5	104.12	44.740	16.10	234.00	101.05
Independent Variables					
IU	4.561	6.519	0.004	34.253	1.549
FTS	1.528	3.307	0.005	22.620	0.504
MCS	28.685	29.752	0.018	138.80	20.008
Control Variables					
GDP	546.1	330.13	111.9	2032.6	467.7
SCH	93.17	29.436	16.63	156.03	92.38
IM	69.38	18.056	16.00	99.00	70.00
IDPT	70.31	19.570	19.00	99.00	74.00
HE	30.242	19.067	4.691	139.75	24.814
ADR	90.09	11.710	56.61	111.94	91.06
SR	1.038	0.014	1.010	1.071	1.030
UP	29.744	11.374	8.246	55.60	29.909
CO2E	0.246	0.497	0.017	3.343	0.0947
Instrumental Variables					
ICC	0.277	0.448	0.000	1.000	0.000
CCSEB	6.988	6.408	0.144	42.219	5.631

Own source: Author calculation.

**Table 2 ijerph-19-07338-t002:** Panel Granger causality test results.

	lnCMU5			lnINFM		
	Ztilde	Zbar	Wbar	Ztilde	Zbar	Wbar
lnIU	8.703 ***	12.19 ***	4.317 ***	8.273 ***	11.618 ***	4.162 ***
lnFTS	12.489 ***	17.225 ***	5.688 ***	13.606 ***	18.711 ***	6.092 ***
lnMCS	39.138 ***	52.674 ***	15.336 ***	35.051 ***	47.237 ***	13.856 ***

Significance level *** *p* < 0.01.

**Table 3 ijerph-19-07338-t003:** FE using Driscoll–Kraay robust error estimation results for infant mortality and child mortality under 5.

	INFM				CMU5			
	1	2	3	4	5	6	7	8
lnIU	−0.070 ***			−0.040 ***	−0.059 ***			−0.051 ***
	0.016			(0.012)	0.011			(0.018)
lnFTS		0.003		0.006		0.001		0.008
		0.013		(0.009)		0.010		(0.013)
lnMCS			−0.059 ***	−0.030 ***			−0.052 ***	−0.031 ***
			0.011	(0.005)			0.009	(0.007)
lnGDP	−0.152 ***	−0.286 ***	−0.124 ***	−0.073 ***	−0.114 ***	−0.226 ***	−0.084 ***	−0.110 ***
	0.027	0.059	0.036	(0.024)	0.024	0.050	0.031	(0.026)
lnSCH	−0.084	−0.139	−0.069	−0.003	−0.025	−0.071**	−0.008	−0.062
	0.071	0.065	0.070	(0.038)	0.038	0.031	0.037	(0.073)
lnIM	0.142 ***	0.197 ***	0.191 ***	0.120 ***	0.110 **	0.157 ***	0.151 ***	0.151 ***
	0.047	0.041	0.040	(0.036)	0.046	0.031	0.028	(0.037)
lnIDPT	−0.167 ***	−0.237 ***	−0.197 ***	−0.120 ***	−0.121 ***	−0.180 ***	−0.144 ***	−0.165 ***
	0.047	0.064	0.058	(0.027)	0.037	0.047	0.039	(0.040)
lnHE	−0.076	−0.142 ***	−0.089 **	−0.071	−0.083 **	−0.138 ***	−0.091 **	−0.064
	0.052	0.044	0.044	(0.044)	0.041	0.036	0.037	(0.055)
lnADR	−0.080	0.075	0.034	0.127	0.105	0.235 *	0.198 *	−0.057
	0.125	0.154	0.139	(0.106)	0.096	0.125	0.111	(0.136)
lnSR	−23.289 ***	−25.727 ***	−26.321 ***	−19.500 ***	−18.768 ***	−20.850 ***	−21.333 ***	−23.955 ***
	5.412	6.082	5.044	(3.190)	4.057	4.589	3.659	(4.291)
lnUP	−0.079 *	−0.148 **	−0.100 **	−0.033	−0.042	−0.100 *	−0.057	−0.069
	0.048	0.068	0.050	(0.038)	0.040	0.058	0.040	(0.045)
lnCO2E	0.039 **	0.064 ***	0.050 ***	0.025**	0.026**	0.048 ***	0.035 ***	0.037 **
	0.015	0.010	0.013	(0.010)	0.011	0.007	0.009	(0.015)
R2	0.850	0.805	0.846	0.859	0.825	0.781	0.817	0.832
Adj. R2	0.838	0.789	0.834	0.847	0.811	0.764	0.803	0.818
F-Stat	254.624 ***	184.781 ***	246.956 ***	227.488 ***	211.733 ***	211.733 ***	211.733 ***	184.226 ***
	Lagrange Multiplier Test (Breusch–Pagan) for balanced panels, X^2^							
	1984.1 ***	1846.2 ***	1984.1 ***	1930.7 ***	1670.5 ***	1603.2 ***	1670.5 ***	1665.3 ***
	F test for individual effects							
	124.83 ***	95.708 ***	131.94 ***	126.42 ***	88.073 ***	71.707 ***	91.224 ***	88.682 ***
	Breusch–Pagan LM test for cross-sectional dependence in panels							
	1485.8 ***	1333.9 ***	1595.3 ***	1609 ***	1381.9 ***	1287.7 ***	1549.1 ***	1565 ***
	Pesaran CD test for cross-sectional dependence in panels							
	1572.8 ***	1380.5 ***	1661.4 ***	7.690 ***	1573.6 ***	1363.9 ***	1646.9 ***	9.518 ***
	Hausman Test							
	111.79 ***	69.428 ***	553.31 ***	242.09 ***	117.69 ***	80.781 ***	222.17 ***	145 ***

Note: Significance level * *p* < 0.1; ** *p* < 0.05; *** *p* < 0.01.

**Table 4 ijerph-19-07338-t004:** Two-way FE using Driscoll–Kraay robust error estimation results for infant mortality and child mortality under 5.

	CMU5				INFM			
	1	2	3	4	5	6	7	8
lnIU	−0.053			−0.012	−0.049			−0.010
	0.047			(0.010)	0.042			(0.006)
lnFTS		−0.007		−0.005		−0.006		−0.004
		0.010		(0.011)		0.007		(0.008)
lnMCS			−0.028 ***	−0.023 ***			−0.023 ***	−0.020 ***
			0.006	(0.007)			0.006	(0.006)
lnGDP	−0.083 *	−0.115 **	−0.092 **	−0.083 **	−0.048	−0.078 *	−0.058 *	−0.051
	0.045	0.046	0.040	(0.039)	0.036	0.040	0.034	(0.034)
lnSCH	−0.042	−0.050	−0.037	−0.038	0.013	0.006	0.017	0.016
	0.075	0.071	0.071	(0.072)	0.041	0.037	0.037	(0.037)
lnIM	0.157 ***	0.221 ***	0.193 ***	0.185 ***	0.126 ***	0.185 ***	0.162 ***	0.155 ***
	0.056	0.039	0.029	(0.035)	0.062	0.040	0.031	(0.038)
lnIDPT	−0.185 ***	−0.234 ***	−0.213 ***	−0.205 ***	−0.135 ***	−0.180 ***	−0.162 ***	−0.156 ***
	0.034	0.044	0.042	(0.038)	0.028	0.026	0.024	(0.023)
lnHE	−0.059	−0.083	−0.075	−0.073	−0.065	−0.087 **	−0.080 **	−0.079 *
	0.051	0.051	0.049	(0.055)	0.044	0.041	0.039	(0.044)
lnADR	−0.091	−0.009	−0.013	−0.031	0.090	0.166	0.163	0.148
	0.108	0.128	0.135	(0.138)	0.091	0.101	0.107	(0.107)
lnSR	−19.318 ***	−19.364 ***	−20.299 ***	−20.315 ***	−15.645 ***	−15.671 ***	−16.456 ***	−16.478 ***
	4.973	4.864	4.570	(4.343)	3.768	3.683	3.462	(3.341)
lnUP	−0.022	−0.027	−0.034	−0.035	0.009	0.005	−0.001	−0.001
	0.024	0.034	0.030	(0.024)	0.017	0.027	0.023	(0.019)
lnCO2E	0.000	−0.003	−0.004	−0.002	−0.002	−0.005	−0.005	−0.004
	0.015	0.008	0.006	(0.008)	0.012	0.005	0.004	(0.006)
R-Squared:	0.261	0.276	0.290	0.296	0.290	0.316	0.333	0.339
Adj. R-Squared:	0.170	0.187	0.203	0.206	0.203	0.232	0.252	0.255
F-statistic on 10 and 432 DF	160.452	164.994	177.230	15.039	190.025	200.290	216.549	18.387

Significance level * *p* < 0.1; ** *p* < 0.05; *** *p* < 0.01.

**Table 5 ijerph-19-07338-t005:** PCCE using Driscoll–Kraay robust error estimation results for infant mortality and child mortality under 5.

	CMU5				INFM			
	1	2	3	4	5	6	7	8
lnIU	−0.003 ***			−0.001	−0.002 ***			−0.001
	(0.001)			(0.001)	(0.001)			(0.001)
lnFTS		0.000		−0.011 **		−0.002		−0.006 **
		(0.002)		(0.004)		(0.003)		(0.002)
lnMCS			0.000	0.002 ***			−0.001	0.000
			(0.001)	(0.001)			(0.001)	(0.001)
lnGDP	−0.001	−0.009	−0.004	0.005	−0.006 **	−0.012 *	−0.005 *	−0.005
	(0.005)	(0.007)	(0.003)	(0.007)	(0.003)	(0.007)	(0.003)	(0.003)
lnSCH	0.002	−0.009	−0.005	−0.023 **	−0.006 *	−0.031 **	−0.009 **	−0.012 *
	(0.004)	(0.006)	(0.004)	(0.009)	(0.003)	(0.012)	(0.004)	(0.006)
lnIM	0.012 ***	0.029 ***	−0.004	−0.026 **	0.005	0.014 **	−0.01 **	−0.016 **
	(0.004)	(0.01)	(0.008)	(0.013)	(0.003)	−0.006	(0.004)	(0.008)
lnIDPT	−0.021 ***	−0.017 **	0.000	0.016 *	−0.013 ***	−0.010	0.006 *	0.013 **
	(0.005)	(0.009)	(0.007)	(0.01)	(0.003)	(0.009)	(0.003)	(0.006)
lnHE	−0.001	−0.015 ***	0.008 ***	0.015 **	0.001	−0.006 **	0.009 ***	0.012 ***
	(0.004)	(0.004)	(0.003)	(0.006)	(0.002)	(0.003)	(0.001)	(0.004)
lnADR	0.115 ***	0.129 ***	−0.105	−0.159 *	0.033	0.071 **	−0.159 ***	−0.169 ***
	(0.026)	(0.025)	(0.074)	(0.087)	(0.023)	(0.03)	(0.04)	(0.04)
lnSR	−0.141	−0.120	−0.007	−2.665 **	0.051	0.068	0.227	−0.888
	(0.794)	(0.933)	(0.736)	(1.076)	(0.618)	(1.154)	(0.58)	(0.928)
lnUP	−0.069	0.125	0.075 **	0.118 **	0.025	0.203 ***	0.136 ***	0.185 ***
	(0.051)	(0.079)	(0.037)	(0.046)	(0.055)	(0.059)	(0.02)	(0.036)
lnCO2E	−0.009 ***	−0.007 ***	−0.009 ***	−0.009 ***	−0.004 ***	−0.006 ***	−0.005 ***	−0.005 ***
	(0.002)	(0.001)	(0.002)	(0.002)	(0.001)	(0.002)	(0.001)	(0.001)
Total Sum of Squares:	131,245	131,245	131,245	131,245	90,734	90,734	90,734	90,734
Residual Sum of Squares:	0.017	0.038	0.013	0.008	0.009	0.023	0.006	0.003
HPY R2:	0.998	0.996	0.999	0.999	0.999	0.997	0.999	0.999

Significance level * *p* < 0.1; ** *p* < 0.05; *** *p* < 0.01.

**Table 6 ijerph-19-07338-t006:** System-GMM estimation results for infant mortality and child mortality under 5.

	INFM			CMU5		
	1	2	3	4	5	6
IU	−0.093 ***			−0.105 ***		
	(0.026)			(0.030)		
FTS		−0.051			−0.046	
		(0.033)			(0.037)	
MCS			−0.036			−0.040
			(0.024)			(0.027)
GDP	−0.170 *	−0.281 **	−0.270 **	−0.136	−0.271 **	−0.249 **
	(0.091)	(0.124)	(0.109)	(0.102)	(0.134)	(0.117)
SCH	−0.132	−0.167	−0.181 *	−0.276 ***	−0.317 ***	−0.331 ***
	(0.097)	(0.102)	(0.095)	(0.102)	(0.104)	(0.100)
IM	−0.292	−0.122	−0.176	−0.370	−0.181	−0.238
	(0.319)	(0.314)	(0.321)	(0.342)	(0.342)	(0.347)
IDPT	0.252	0.118	0.165	0.313	0.159	0.213
	(0.316)	(0.327)	(0.330)	(0.332)	(0.352)	(0.349)
HE	0.115	−0.001	0.070	0.127	0.003	0.076
	(0.084)	(0.105)	(0.101)	(0.091)	(0.112)	(0.111)
ADR	1.176 ***	1.104 ***	1.176 ***	1.172 ***	1.468 ***	1.382 ***
	(0.096)	(0.110)	(0.096)	(0.102)	(0.106)	(0.108)
SR	−2.547	−3.824 *	−2.547	−6.324 ***	−5.040 *	−6.285 ***
	(2.576)	(2.221)	(2.576)	(2.030)	(2.718)	(2.214)
UP	0.395 ***	0.388 ***	0.431 ***	0.333 ***	0.334 ***	0.374 ***
	(0.075)	(0.101)	(0.088)	(0.088)	(0.110)	(0.098)
CO2E	−0.136 **	−0.054	−0.114 **	−0.155 **	−0.070	−0.130 **
	(0.064)	(0.044)	(0.056)	(0.068)	(0.049)	(0.060)
Sargan test: chisq (162) =	27	27	27	27	27	27
*p*-value =	1	1	1	1	1	1
Autocorrelation test (1): normal =	−1.230	−0.615	−0.957	−1.595	−1.168	−1.405
*p*-value =	0.219	0.538	0.338	0.111	0.243	0.160
Autocorrelation test (2): normal =	−0.432	−0.187	−0.410	−0.693	−0.361	−0.529
*p*-value =	0.665	0.852	0.682	0.489	0.718	0.597
Wald test for coefficients: chisq (10) =	37,675.47	25,405.61	27,954.66	63,688.66	27,759.88	32,799.97
*p*-value =	0.000	0.000	0.000	0.000	0.000	0.000

Note: Significance level * *p* < 0.1; ** *p* < 0.05; *** *p* < 0.01.

## Data Availability

The datasets used and/or analyzed during the current study are available from the corresponding author on reasonable request.

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
