# Peer review of "Information Communication Technology and Infant Mortality in Low-Income Countries: Empirical Study Using Panel Data Models"

_ijerph, 2022, doi:10.3390/ijerph19127338_

Round 1

Reviewer 1 Report

The authors aim to investigate the 
 the causal effect of ICT on Infant mortality using panel data in low and middle-income countries. The manuscript is well written and will be of great interest to the reader. I recommend getting this paper published in a professional journal. However, I suggest authors to check for typos and other grammatical errors before it is being accepted.

Author Response

Reviewer1

Comments and Suggestions for Authors

The authors aim to investigate the causal effect of ICT on Infant mortality using panel data in low and middle-income countries. The manuscript is well written and will be of great interest to the reader. I recommend getting this paper published in a professional journal. However, I suggest authors to check for typos and other grammatical errors before it is being accepted.

Response: The authors appreciate the comments provided by the honored reviewer. In this version the authors have carefully enhanced the papers concerns of all the referees. And to address the typos and grammatical errors of the manuscript through MDPI editing services. Therefore we thank you for your valuable comments and guidance and the time to review our manuscript.

Reviewer 2 Report

In this paper, the authors study the influence of three ICT on health outcomes, considering data from multiple low-income countries. In my opinion, the current version of the manuscript is not suitable for publication. I think that a deeper revision of the text is still needed in order to improve the grammar and the quality of tables, figures, and equations. Besides, the authors provide too many details about the econometric models fitted, considering that the manuscript has been submitted to a public health journal. These are important but make the paper very difficult to read in my opinion, especially for the potential audience of the journal. Maybe the authors should consider providing some of these technical details as an Appendix.

Author Response

Comments and Suggestions for Authors

General comment: In this paper, the authors study the influence of three ICT on health outcomes, considering data from multiple low-income countries. In my opinion, the current version of the manuscript is not suitable for publication. I think that a deeper revision of the text is still needed in order to improve the grammar and the quality of tables, figures, and equations. Besides, the authors provide too many details about the econometric models fitted, considering that the manuscript has been submitted to a public health journal. These are important but make the paper very difficult to read in my opinion, especially for the potential audience of the journal. Maybe the authors should consider providing some of these technical details as an Appendix.

Response:

Response: The authors appreciate the comments provided by the honored reviewer. In this version the authors have addressed the reviewer concerns through:

  1. We have sent the manuscript for the MDPI English editing service (the journal English editing services) to deal with text inconsistencies, English grammar typos and general grammar.
  2. As per tables, graphs and equations, we have followed the latest published papers in the journal and revised the equations and tables forms accordingly. As per graphs and figures, we believe that in the editorial process of copying the figures were damaged, yet we have changed them with the original ones which are clearer.
  3. We have reduced some of the tables in both results and econometric models to the appendix to make it more compact. We also put the tests definitions and use reasons in the appendix part as we believe the test have in depth technical information that also might not be suitable for the reader of the journal. Yet, if there is a specific part that the referee believes could be moved to the appendix, we would appreciate that it is highlighted or pointed so that we move it to the appendix part. Especially that we follow the guidance of the esteemed referee and view point of addressing the potential audience of the journal.

We thank the referee for taking time and effort to review our manuscript.

Reviewer 3 Report

Khelfaoui and colleagues submit a manuscript on the use of panel data models on World Bank data to evaluate the possible impact of information communication technologies on infant mortality in low-income countries. This is necessarily a complex study, and the resulting manuscript is expected to be long and to contain much detail. However, the present version of the paper is too long, not well organized, and difficult to read. Details on these general impressions follow below.

The written English used in the paper is difficult to understand, and some expressions seem to be derived from a machine translation engine.  It is absolutely essential that this paper undergo substantial revisions with the aid of a native English speaker, not just to clarify sentences with uncertain meaning, awkward grammar and numerous orthographic errors, but also because some terminology used may actually be misleading.

Just a few examples of awkward terminology include:

In the abstract: “… ICT plays a promising tool to elevate this issue."; "need to show a more adhering role to facilitate the diffusion…"

In the text: “… if health is infected anywhere, it will affect everywhere.";  on page 3, "admittance to significant healthcare data", did the authors mean “access to…”??  In the last paragraph on page 3, is "acquiescence" intended to mean acquisition?  On page 4, "pointers of wellbeing results" is probably intended to read "indicators of wellbeing".  In the last line of page 4, "comprehensive" is probably intended to mean "generalized".

Some terminology that is probably incorrect, or has uncertain meaning: In the introduction, third paragraph, "unemployment elevation" was probably intended to state decrease in unemployment (or increase in employment); fifth paragraph, “1 Separate and different from the global scope, results and findings…”.  On page 4, "… high anxiety is negatively associated with content towards doctors' consolation…” - the meaning of this sentence is incomprehensible.

Structurally, the paper could be substantially improved by significant reorganization of contents.  The section entitled "A short theoretical literature [sic: ?review/discussion?] of the relationship between health and ICT" reads as if it is part of a thesis dissertation, and it should be a trimmed substantially for a Journal manuscript; some essential parts can be moved to the Introduction, others to the Discussion section; alternatively, this extension to the introduction might be added as an appendix to the main paper.

The Methods section is a long and detailed, but it is easier to understand than other parts of the text from a linguistic perspective; it is also well referenced. Of note, although the authors’ use of Infant Mortality is standard in the field, the term “Infant Mortality Under 5” is confusing, since it usually expressed as “Child Mortality under 5”; the IMU5 term is not standard in the field. Methodologically, the variable Child Mortality under 5 overlaps with, and includes all the information contained in the Infant Mortality variable; therefore these 2 dependent variables should be highly correlated. Did the authors consider studying effects of ICT on infant mortality (<1 year) and mortality for the 1-5 age cohort separately, since these 2 separate age groups might reveal different effects of ICT on health?

The Results are reported logically, in substantial detail. Although some graphs and tables could be moved to an Appendix to make the body of the paper more compact and easier to follow for more casual readers, the current sequence of information displayed under Results may be advantageous for researchers who are heavily involved in this topic and use similar methods.

Note that this reviewer would be unable to spot methodological errors in the more sophisticated statistical analysis used in this work.

The Conclusions section is too long, and much of the first paragraph could be merged into the Discussion, as needed. The implications and policy suggestions are appropriate based on the results.

References in the text and bibliography do not follow the formatting guidelines for the Journal.  It appears that the Majeed and Khan 2019 reference is duplicated.

Author Response

Reviewer 3

Comments and Suggestions for Authors

General comment: Khelfaoui and colleagues submit a manuscript on the use of panel data models on World Bank data to evaluate the possible impact of information communication technologies on infant mortality in low-income countries. This is necessarily a complex study, and the resulting manuscript is expected to be long and to contain much detail. However, the present version of the paper is too long, not well organized, and difficult to read. Details on these general impressions follow below.

 Response: The authors appreciate the comments provided by the honored reviewer. In this version the authors have gone through the manuscript and responded on a point-by-point comment in order to reduce the length and organize the paper. We also tried our best to utilize appropriate ways to make it readable.

  1. Comment: The written English used in the paper is difficult to understand, and some expressions seem to be derived from a machine translation engine.  It is absolutely essential that this paper undergo substantial revisions with the aid of a native English speaker, not just to clarify sentences with uncertain meaning, awkward grammar and numerous orthographic errors, but also because some terminology used may actually be misleading.

Just a few examples of awkward terminology include:

In the abstract: “… ICT plays a promising tool to elevate this issue."; "need to show a more adhering role to facilitate the diffusion…"

In the text: “… if health is infected anywhere, it will affect everywhere.";  on page 3, "admittance to significant healthcare data", did the authors mean “access to…”??  In the last paragraph on page 3, is "acquiescence" intended to mean acquisition?  On page 4, "pointers of wellbeing results" is probably intended to read "indicators of wellbeing".  In the last line of page 4, "comprehensive" is probably intended to mean "generalized".

Response: The authors appreciate the comments provided by the honored reviewer. In this version the authors have addressed this comment through utilizing the English editing service of the journal therefore we hope that the resulting editing is up to the standards.

  1. Comment: Some terminology that is probably incorrect, or has uncertain meaning: In the introduction, third paragraph, "unemployment elevation" was probably intended to state decrease in unemployment (or increase in employment); fifth paragraph, “1 Separate and different from the global scope, results and findings…”.  On page 4, "… high anxiety is negatively associated with content towards doctors' consolation…” - the meaning of this sentence is incomprehensible.

Response: The authors appreciate the comments provided by the honored reviewer. Similarly, to the previous comment in this version the authors have utilized the journal English service to remedy the terminotics and English issues. For example in the new version:

"unemployment elevation" is now “unemployment reduction”

“Separate and different from the global scope, results and findings…” is now “To separate and differentiate our study from the global scope as well as the global results and findings”

  1. Comment: Structurally, the paper could be substantially improved by significant reorganization of contents.  The section entitled "A short theoretical literature [sic: ?review/discussion?]of the relationship between health and ICT" reads as if it is part of a thesis dissertation, and it should be a trimmed substantially for a Journal manuscript; some essential parts can be moved to the Introduction, others to the Discussion section; alternatively, this extension to the introduction might be added as an appendix to the main paper.

Response: The authors appreciate the comments provided by the honored reviewer. In this version the authors have addressed this comment and have entitled the section “Review of the relationship between health and ICT”

  1. Comment: The Methods section is a long and detailed, but it is easier to understand than other parts of the text from a linguistic perspective; it is also well referenced. Of note, although the authors’ use of Infant Mortality is standard in the field, the term “Infant Mortality Under 5” is confusing, since it usually expressed as “Child Mortality under 5”; the IMU5 term is not standard in the field. Methodologically, the variable Child Mortality under 5 overlaps with, and includes all the information contained in the Infant Mortality variable; therefore these 2 dependent variables should be highly correlated. Did the authors consider studying effects of ICT on infant mortality (<1 year) and mortality for the 1-5 age cohort separately, since these 2 separate age groups might reveal different effects of ICT on health?

Response: The authors appreciate the comments provided by the honored reviewer. To reduce the length of the complicated details in the methodology part, in this version, the authors have moved the first table to the appendix part for to make this part more concise. We also believe thank you for the guidance where both you and the second referee put a note on moving some of the econometric in-depth details. In this aim, we moved the most in-depth part which are the econometric tests to the appendix as well. Hence, if there are other parts that the referees seem too technical and also need to be moved to appendix, the authors would willingly follow the honorable guidance of therefrees. Of note, we, the authors’ used Infant Mortality, in this version, we changed it to the usual expression in the field, “Child Mortality under 5 CMU5”. Methodologically, the variable Child Mortality under 5 overlaps with, and includes all the information contained in the Infant Mortality variable; therefore these 2 dependent variables should be highly correlated. And to remove the confusion of them highly correlated we have made their correlation matrices separately. As we do believe, that they overlap, yet in the variable selection process, as illustrated in our data processing part, we only utilized the variables that have been previously mentioned articles and research works, in order to have a comparative ground.

  1. Comment: The Results are reported logically, in substantial detail. Although some graphs and tables could be moved to an Appendix to make the body of the paper more compact and easier to follow for more casual readers, the current sequence of information displayed under Results may be advantageous for researchers who are heavily involved in this topic and use similar methods.

Response: The authors appreciate the comments provided by the honored reviewer. In this version and to reduce the length of the results part, the authors have moved the panel unit roots table, instrumental variables tables to and correlation matrices graphs to the appendix.

  1. Comment: Note that this reviewer would be unable to spot methodological errors in the more sophisticated statistical analysis used in this work.

Response: The authors appreciate the comments provided by the honored reviewer. In this version the authors have addressed this issue through reducing the length of the methodological part. We have put the difficult technical parts in the appendix and kept the easier to understand parts by the target reader of the journal. We have moved the table of variable descriptions for its lengthy and moved the tests definitions to the appendix as well.

  1. Comment: The Conclusions section is too long, and much of the first paragraph could be merged into the Discussion, as needed. The implications and policy suggestions are appropriate based on the results.

Response: The authors appreciate the comments provided by the honored reviewer. In this version the authors have addressed the length of the conclusion to reduce it to the journal average conclusion length of other articles. And we appreciate that the reviewer believes that the implication and policy suggestions are appropriate base on the results.

The new conclusion is as follows:

“To conclude, our study expanded the current literature on health factors using an empirical analysis of ICT on health outcomes for 27 selected low-income countries for 18 years. We applied suitable panel estimation approaches to address potential error hetero-geneity and cross-sectional dependencies (i.e., Driscoll–Kraay errors and PCCE). The health-outcome-dependent variables of this study were INFM and CMU5. The independ-ent ICT proxies were  IU,  FTS, and MCS. In this study, we observed a strong presence of cross-sectional dependency in the models and data. Our panel data were stationary through both generations of panel unit-root tests. We established causality through panel Granger causality tests. In the FE results, we found that both IU and MCS decreased the rates of INFM and CMU5. In almost all the estimation results, we found that IU had a significant effect in the reduction in INFM and CMU5 rates. In the joint model of PCCE, fixed telephones may have also reduced both mortality rates while mobile devices appeared to increase them. Finally, due to the setting of our data and the application of novel models, our results differed from previous research and indicated that findings on a global scale were different to those found in low-income countries.

Our conclusions also provided policy suggestions for consideration. Policymakers and leaders of low-income countries should implement policies that guarantee sufficient internet access for the community. The internet, as a means of information transfer and a tool for communication, could improve these countries' health outcomes by promoting health education, rapid health information sharing for early discovery and prevention of diseases, and healthcare systems support as well as overcoming the barriers of location and time for communication and consultation between patients and healthcare providers. Finally, and most importantly, the internet and mobile devices serve as great tools to in-form and educate healthy behaviors, particularly for pregnant people, to reduce the child mortality rates through early contact with specialist physicians and healthcare systems, and to provide support and understanding for pregnancy and early childhood issues through the exchange of information.”

  1. Comment: References in the text and bibliography do not follow the formatting guidelines for the Journal.  It appears that the Majeed and Khan 2019 reference is duplicated.

Response: The authors appreciate the comments provided by the honored reviewer. In this version the authors have utilized Zotero application to adhere to the referencing style and guidelines of the journal. And in the manuscript body, the authors have unified the mentioning of the authors where it’s needed.

Round 2

Reviewer 2 Report

I think that the authors have improved the paper substantially. Therefore, I am in favor of the publication.

Reviewer 3 Report

The authors adequately addressed the concerns from the reviewers about the contents of the paper, and moving some elements to the Appendix helped the readability of the paper. However, the manuscript remains quite long, which will be a disincentive for the average reader who is interested in the topic; it is likely that only researchers who are deeply involved in this subject matter and methodology will commit to reading through the paper.